# Carbonic Anhydrase IX: A Renewed Target for Cancer Immunotherapy

**DOI:** 10.3390/cancers14061392

**Published:** 2022-03-09

**Authors:** Najla Santos Pacheco de Campos, Bruna Santos Souza, Giselle Correia Próspero da Silva, Victoria Alves Porto, Ghanbar Mahmoodi Chalbatani, Gabriela Lagreca, Bassam Janji, Eloah Rabello Suarez

**Affiliations:** 1Center for Natural and Human Sciences, Federal University of ABC, Santo André 09210-580, SP, Brazil; naj.pacheco@gmail.com (N.S.P.C.); bruna.santos.souza00@gmail.com (B.S.S.); giselle.correia97@gmail.com (G.C.P.S.); victoria.porto@aluno.ufabc.edu.br (V.A.P.); gabrielalagreca@gmail.com (G.L.); 2Tumor Immunotherapy and Microenvironment (TIME) Group, Department of Cancer Research, Luxembourg Institute of Health, 1445 Luxembourg, Luxembourg; ghanbar.mahmoodichalbatani@lih.lu

**Keywords:** chimeric antigen receptor, antitumor monoclonal antibodies, clear cell renal cell cancer, hypoxic tumors, immunotherapies, immune checkpoint inhibitors, carbonic anhydrase

## Abstract

**Simple Summary:**

Carbonic anhydrase IX (CAIX) has been explored for a long time as a therapeutic target in the fight against clear cell renal cell carcinoma and several hypoxic tumors, usually offering modest results followed by adverse effects. However, recent studies using different antibodies and adoptive cell therapies against CAIX have generated exciting prospects for the immunotherapy of these tumors. This complete review will approach the past and future of anti-CAIX immunotherapies.

**Abstract:**

The carbonic anhydrase isoform IX (CAIX) enzyme is constitutively overexpressed in the vast majority of clear cell renal cell carcinoma (ccRCC) and can also be induced in hypoxic microenvironments, a major hallmark of most solid tumors. CAIX expression is restricted to a few sites in healthy tissues, positioning this molecule as a strategic target for cancer immunotherapy. In this review, we summarized preclinical and clinical data of immunotherapeutic strategies based on monoclonal antibodies (mAbs), fusion proteins, chimeric antigen receptor (CAR) T, and NK cells targeting CAIX against different types of solid malignant tumors, alone or in combination with radionuclides, cytokines, cytotoxic agents, tyrosine kinase inhibitors, or immune checkpoint blockade. Most clinical studies targeting CAIX for immunotherapy were performed using G250 mAb-based antibodies or CAR T cells, developed primarily for bioimaging purposes, with a limited clinical response for ccRCC. Other anti-CAIX mAbs, CAR T, and NK cells developed with therapeutic intent presented herein offered outstanding preclinical results, justifying further exploration in the clinical setting.

## 1. Introduction

Carbonic anhydrases are metalloenzymes that reversibly catalyze the hydration of carbon dioxide, generating bicarbonate ions and protons [1]. Several tumors, such as clear cell renal cell carcinoma (ccRCC), glioblastoma, triple-negative breast cancer, ovarian cancer, colorectal, and others [2] overexpress carbonic anhydrase isoform IX (CAIX). This transmembrane enzyme differs from most other CAs by having its catalytic site located in the extracellular domain, responsible for tumor microenvironment acidification [1]. In consequence of the low pH, cathepsin B and other proteolytic enzymes are activated, creating a favorable environment for cancer cell migration and metastasis. An acidic pH also impairs the tumoricidal function of cytotoxic T cells and natural killer cells (NK), favoring the occurrence of minimal residual disease and recurrence [3].

CAIX expression occurs when tumor growth exceeds vascularization due to hypoxia. In this condition, the inhibition of an enzyme called prolyl-hydroxylase occurs since this enzyme uses oxygen as a co-substrate, resulting in a dissociation between the hypoxia-inducible factor 1α (HIF-1α) and von Hippel Lindau (pVHL) protein. This process results in HIF-1α accumulation and subsequent dimerization with HIF-1β, activating the transcription of several hypoxia response genes, including CAIX [4]. A mutation in the pVHL-coding gene present in about 95% of clear cell renal carcinoma (ccRCC) cases can also be responsible for HIF-1α accumulation, leading to the CAIX constitutive expression found in this cancer type [5,6,7]. In addition to tumors, CAIX expression is restricted to a few healthy tissues, such as intrahepatic biliary ducts, gastric mucosa, and duodenum [8], highlighting its potential for developing cancer-targeted therapies.

Immunotherapy with monoclonal antibodies has emerged in the last decades as a modality of cancer treatment with less toxicity when compared to conventional chemotherapy and radiotherapy treatments, increasing the survival rate for several patients. More recently, adoptive cell therapies, especially those driving T cells or NK cells against the tumor using the expression of chimeric antigen receptors (CAR) against tumor-associated antigens (TAAs), are being positioned as powerful strategies against cancer. The CAR acts independently of the expression of antigens via MHC for T cell activation, and neither needs an external co-stimulatory signal, transposing several mechanisms of tumor immune evasion [9,10]. The following sections will provide a summary of preclinical and clinical results of immunotherapeutic strategies based on monoclonal antibodies (mAbs), fusion proteins, and CAR T or NK cells targeting CAIX against different types of solid malignant tumors.

## 2. Anti-CAIX Monoclonal Antibodies: Preclinical and Clinical Efficacy

This section will present antitumor responses and adverse effects of different anti-CAIX mAbs available, used alone or in combination with either radioisotopes or cytokines. Table 1 and Table 2 summarize chronologically the primary data of preclinical and clinical studies based on anti-CAIX mAbs, respectively.

### 2.1. Murine G250 IgG1 mAb—Isolated and Associated with Radionuclides

Murine G250. IgG1 mAb (mG250) was one of the first anti-CAIX antibodies developed and tested for ccRCC detection and treatment. Preclinical studies performed in vivo and ex vivo in perfusion kidneys containing ccRCC and clinical trials have shown the potential of the molecule as a bioimaging agent, conjugated with ^99m^Tc, ^125^I, or ^131^I-mG250 antibodies [11,12,13]. Phase I and II clinical trials with ^131^I-mG250 using different doses of ^131^I and 10 mg of G250 in a single dose injection at doses greater than 30 mCi/m^2^ induced important hematotoxicity and hepatotoxicity. The maximum tolerated dose (MTD) of 90 mCi/m^2^ was used in 45% of the patients (15/33). Two patients had a 30–35% reduction in the sum of the diameters in lung metastases without new injuries, and 51% presented stable disease. However, all patients developed human antimouse antibodies (HAMA) within four weeks, excluding the possibility of retreatment [14]. Radioimmunotherapy using two other radionuclides (^111^In e ^177^Lu) conjugated to mG250 was also tested against human ccRCC xenografts in mice. Treatment with ^177^Lu-benzyl-isothiocyanate-1,4,7,10-tetraazacyclododecane-tetraacetic acid (DOTA)-mG250 almost tripled the median survival when compared to ^111^In-DOTA-mG250 and ^177^Lu-DOTA conjugated with an unspecific antibody, demonstrating the superior performance of the radionuclide Lutetium 177 conjugated with mG250 for the treatment of human ccRCC xenografts [15]. The mG250 mAb without radioisotope conjugation had its efficacy tested for treating human colorectal carcinoma cells (HT-29) in a murine subcutaneous model. In this study, one of the groups treated with mG250 injected ten days after tumor implantation responded with three-fourths tumor volume shrinkage compared to the control group [16].

**Table 1 cancers-14-01392-t001:** Anti-CAIX monoclonal antibodies-based preclinical studies reporting antitumor responses.

Author	Antibody Type	Tumor Type	Dosage	Response
Surfus et al. (1996) [17]	cG250	RCC and breast carcinoma cell lines	cG250: 0.5 µg/mL, IL2: 100 U/mL	ADCC with PBMCs (effector to target rate 100:1) after 4 h RCC—SK-RC-13: cG250 48%; cG250 + IL2 50%; SK-RC-30: cG250 25%; cG250 + IL2 65%; Breast cancer—BT-20: cG250 38%; cG250 + IL2 28%
Liu et al. (2002) [18]	cG250	RCC and chronic myelogenous leukemia	cG250: 1 µg/mL, IL2 10 IU/mL; IFNγ, IFN-2a, IFN-2b 1000 IU/mL	ADCC with PBMC (effector to target rate 25:1) after 2 days RCC—SK-RC-52: cG250 + IL2 42%; cG250 + IFN-𝛾 33%; cG250 + IFN-𝛼-2a or cG250 + IFN-α-2b 25%; SK-RC-09: cG250 + IL2 28%; cG250 + IFN-𝛾; cG250 + IFN-𝛼-2a, and cG250 + IFN-α-2b < 10%; Leukemia—K562: cG250 + IL2 60%; cG250 + IFN-𝛾 30%; cG250 + IFN-𝛼-2a or cG250 + INF-α-2b 43%
Brouwers et al. (2004) [19]	^131^I-cG250, ^90^Y-SCN-Bz- DTPA-cG250, ^177^Lu-SCN-Bz-DTPA-cG250, or ^186^Re-MAG3 cG250	RCC	30 µg ^131^I-cG250, 30 µg ^90^Y-SCN-Bz-DTPA-cG250, 60 µg ^177^Lu-SCN-Bz-DTPA-cG250, or 35 µg ^186^Re-MAG3-cG250; Variable doses of radioisotopes	Best median survival (SK-RC-52 cells) ^177^Lu-SCN-Bz-DTPA cG250: 294 days; ^90^Y-SCN-Bz-DTPA cG250: 241 days; ^186^Re-MAG3-cG250: 211 days; ^131^I-cG250: 164 days; Control groups < 150 days
Bauer et al. (2009) [20]	cG250-TNF and cG250	RCC	100 µg of cG250 or cG250-TNF 300 ng every 3 days	In vivo tumor size after 78 days (SK-RC-52 cells) cG250-TNF + IFNγ: 60% decrease; cG250-TNF: 50% decrease; cG250 + IFNγ: no difference in tumor size compared to negative control
Zatovicova et al. (2010) [21]	VII/20	Colorectal carcinoma	100 μg twice a week	In vivo tumor weight/volume reduction (HT-29 cells) 60%/73% treatment initiated after 10 days of tumor implantation; 88%/93% treatment initiated in the same day of tumor implantation
Oosterwijk-Wakka et al. (2011) [22]	^125^I-cG250 + sorafenib, sunitinib, or vandetanib	RCC	^125^I-cG250 185 kBq/5 μg 35 mg/kg of sunitinib, 50 mg/kg of sorafenib, 50 mg/kg of vandetanib	In vivo tumor volume (NU-12 cells) decrease for continuous treatment (14 days) Vandetanib: 57%, sunitinib: 49%, and sorafenib: 37%, all compared to ^125^I-cG250 alone
Petrul et al. (2012) [23]	BAY 79-4620	Colorectal cancer, gastric carcinoma, and NSCLC-PDX	Variable	In vivo tumor regression (3 doses of every 4 days) Colorectal cancer (dose 10 mg/kg): HT-29: 100%, Colo205: 85%; Gastric carcinoma (dose 60 mg/kg): NCI-N87: 87%, MKN-45: 90%, SNU-16: 75%; NSCLC-PDX: complete regression in 2/5, partial regression in 3/5
Muselaers et al. (2014) [15]	^111^In-DOTA-mG250 and ^177^Lu-DOTA-mG250	RCC	13 MBq ^177^Lu-DOTA-mG250, 13 MBq nonspecific ^177^Lu-DOTA-MOPC21, 20 MBq ^111^In-DOTA-mG250	Median survival (SK-RC-52 cells) ^177^Lu-DOTA-mG250: 139 days; ^177^Lu-DOTA-MOPC21: 49 days; ^111^In-DOTA-mG250: 53 days; Control: 49–53 days
Zatovicova et al. (2014) [16]	mG250	Colorectal carcinoma	100 µg/dose	In vivo tumor weight/volume reduction (HT-29 cells) Treatment initiated after 10 days of tumor implantation: 55%/73%; Treatment initiated at the same day of tumor implantation: 90%/93%
Chang et al. (2015) [24]	In vitro: G10, G36, G37, G39, and G119; In vivo: only G37 and G119 were tested	RCC	ADCC in vitro: 5 µg/mL, In vivo: 10 mg/kg	ADCC in SK-RC-09 cells: 25:1 effector to target cells: 25% for G36 and G119; 15–20% for G10, G37, and G39; 50:1 effector to target cells: 45% for G10, G36, G37, and G119; 30% for G39 In vivo tumor weight (Day 29)/volume (Day 28) reduction (SK-RC-59 CAIX+ cells): 85%/75% for G37, G119, mG37, and mG119
Oosterwijk-Wakka et al. (2015) [25]	^111^In-cG250 and Sunitinib	RCC	0.4 MBq/5 µg ^111^In-cG250 three days after administration of 40–50 mg/kg of sunitinib for 13 days	In vivo tumor growth reduction 20 days after the beginning of the treatment with sunitinib NU-12: 60%; SK-RC-52: not statistically significant compared to control
Yamaguchi et al. (2015) [26]	chKM4927 and chKM4927_N297D	RCC	10 mg/kg i.p. twice a week for three weeks	In vivo tumor volume (VMRC-RCW cells) reduction after 32 days chKM4927 and chKM4927_N297D: 60% compared to negative control
Lin et al. (2017) [27]	Anti-CAIX functionalized liposomes with TPL	Lung cancer cells	0.15 mg/kg once every 3–4 days for 8 times via pulmonary delivery	Median survival time (A549 cells) CAIX-TPL-Lips: 90 days (statistically significant compared to saline control); Nontargeted TPL-lips: 71 days (not statistically significant compared to saline control); Control group: 45 days
De Luca et al. (2019) [28]	IL2-Anti-CAIX(XE114)-TNFmut and IL2-Anti-CAIX(F8)-TNFmut	Colon Carcinoma	30 µg i.v. four times every 24 h	Tumor volume reduction (CT26-CAIX cells) after 18 days IL2-F8-TNFmut: 58%; mIL2-F8-mTNFmut: 72%; IL2-XE114-TNFmut: 63%; mIL2-XE114-mTNFmut: 50%

ADCC: antibody-dependent cell cytotoxicity, Bz: benzyl, DOTA: 1,4,7,10-tetraazacyclododecane-tetraacetic acid, DTPA: diethylenetriaminepentaacetic acid, I: iodine, IL2: interleukin-2, In: indium, IFN: interferon, Lu: lutetium, MAG3: mercaptoacetyltriglycine, MOPC21: unspecific control antibody, NSCLC-PDX: non-small cell lung cancer patient-derived xenograft, PBMCs: peripheral blood mononuclear cells, RCC: renal cell cancer, Re: rhenium, TNF: tumor necrosis factor, TNFmut: low potency mutated tumor necrosis factor, TPL: triptolide, Y: yttrium.

**Table 2 cancers-14-01392-t002:** Anti-CAIX monoclonal antibodies-based clinical trials reporting antitumor responses and adverse effects on renal cell cancer.

Author	Phase	Treatment	Clinical Response	Adverse Effects (≥3 Grade)
Divgi et al. (1998) [14]	I/II	mG250 (**10** mg single i.v. infusion) combined with ^131^I (30, 45, 60, 75, and **90** mCi/m^2^)	1/33 CR; 17/33 SD—2 months after treatment	19/33 grade 3 (thrombocytopenia, hematotoxicity, hepatoxicity); 3/33 grade 4 (thrombocytopenia and hematotoxicity); 33/33 HAMA
Steffens et al. (1999) [29]	I	cG250 (**5** mg single i.v. infusion) combined with ^131^l (222–2775 MBq/m^2^)	6/12 PD; 1/12 SD—lasting 3–6 months; 1/12 PR—9 months or longer	1/12 grade 3 (leukocytopenia); 2/12 grade 4 (thrombocytopenia and leukocytopenia); 1/12 HACA
Bleumer et al. (2004) [30]	II	cG250 (**25** mg/m^2^ weekly i.v. infusion for 12 weeks)	10/36 SD, 17/36 PD—week 16; 8/36 SD—week 24; 1/36 CR, 1/36 PR—week 38–44	* 33/36 grade 3 (pain, pulmonary, cardiovascular, constitutional symptoms, neurological, bone marrow, genitourinary, hemorrhage, hepatic, metabolic/laboratory); 5/36 grade 4 (pulmonary, hemorrhage)
Bleumer et al. (2006) [31]	III	cG250 (**20** mg by i.v. infusion for 11 weeks) combined with IL2 (1.8–5.4 MIU daily for 12 consecutive weeks)	1/35 PR, 11/35 SD, 23/35 PD—week 16; 1/35 PR, 7/35 SD, 4/35 PD—week 22	17/35 grade 3 (constitutional symptoms, pain, pulmonary, blood/bone marrow, hepatic); 2/35 grade 4 (renal/genitourinary and metabolic/laboratory); 2/36 HACA
Davis et al. (2007) [32]	Pilot	cG250 (**10** mg/m^2^/week, first and fifth doses trace-labeled with ^131^I) and 1.25 × 10^6^ IU/m^2^/day IL2 for six weeks	2/9 SD, 7/9 PD—after six-week cycle 1; 1/9 SD, 1/9 PD—after six-week cycle 2	* 3/9 grade 3 or 4 (dyspnea and anemia)
Davis et al. (2007) [33]	I	cG250 (**5**, 10, 25, or 50 mg/m^2^ i.v. for 6 weeks) combined with ^131^l (200–350 MBq/m^2^) weeks 1 and 5	1/13 CR, 8/13 SD, 3/13 PD—first six-weeks cycle; 1/13 CR, 6/13 SD, 2/13 PD—second six-weeks cycle	* 1/13 grade 3 (bone pain), 1/13 HACA
Siebels et al. (2010) [34]	I/II	cG250 (**20** mg i.v. infusion; week 2–12) combined with LD-IFNα (3 MIU s.c. 3 times/week; weeks 1–12)	2/26 PR, 14/26 SD—week 16; 1/26 CR, 9/26 SD—24 weeks or longer	11/26 grade 3 (constitutional symptoms, pain, pulmonary, musculoskeletal, cardiovascular, secondary malignancy, lymphatics); 1/26 grade 4 (gastrointestinal)
Stillebroer et al. (2013) [35]	I	cG250 (**10** mg i.v. infusion—three consecutive) combined with ^131^ln (1110–2405 MBq/m^2^)	17/23 SD—during the 3 months 1/23 PR—lasted 9 months	3/23 grade 4 (myelotoxicity); 4/23 HACA
Muselaers et al. (2016) [36]	II	cG250 (**10** mg i.v. infusion) combined with ^111^In (185 MBq/m^2^); ^177^Lu (2405 MBq/m^2^) 9–10 days after infusion; ^177^Lu (1805 MBq/m^2^) weeks 12–14	1/14 PR, 8/14 SD, 5/9 PD—after cycle 1; 1/14 PR, 4/14 SD, 1/14 PD—after cycle 2	12/14 grade 3–4 (thrombocytopenia); 9/14 grade 3–4 (leukocytopenia); 2/14 grade 3 (fatigue and anorexia); 4/14 grade 4 (neutropenia)
Chamie et al. (2017) [37]	III	cG250 (**50** mg i.v.; week 1; 20 mg i.v. weeks 2–24)	NR	72/864 grade 3 or 4—type not mentioned

PD: progressive disease, SD: stable disease, PR: partial response, CR: complete response, MTD: maximum tolerated dose, ND: not detected, NE: not evaluable, NR: no response, HAMA: human antimouse antibodies, HACA: human anti-chimeric antibodies. * All grade 3 and 4 toxicities were not related to the study medication. Doses highlighted in **bold** are related to clinical responses reported.

### 2.2. Humanized Chimeric Monoclonal Antibody IgG1 G250 (cG250)—Isolated or Associated with Cytokines

Due to mG250 toxicity, this antibody was adapted to an IgG1 chimeric humanized version using the variable region of the murine monoclonal antibody G250, being called cG250, WX-G250, or girentuximab (Rencarex^®^, Heidelberg Pharma AG, Ladenburg, Germany). Initial preclinical studies showed that the cG250 antibody could induce cytotoxicity in CAIX-positive cells [18]. In a clinical trial, 36 RCC patients received 50 mg of cG250 (12 infusions, equivalent to 25 mg/m^2^), without the development of human anti-chimeric antibodies (HACA) and with a poststudy median survival of 15 months, with two late clinical responses [30]. Most patients treated in this study developed other types of grade 3 adverse effects (AE), with a few grade 4 cases. A phase III clinical trial evaluating disease-free survival and overall survival in 433 patients treated with cG250 compared to 431 treated with placebo found no significant difference between the groups [37]. Davis et al. (2007) demonstrated a significant decrease in grade 3 or 4 AEs rate using 5 mg/m^2^ cG250 combined with ^131^l to treat patients with metastatic ccRCC or those presenting tumors not eligible for surgical resection [32].

The association of cG250 with interleukin-2 (IL2) in preclinical studies induced relevant antibody-mediated cytotoxicity (ADCC) in RCC and leukemia [17,18]. A clinical trial including 35 ccRCC patients treated with cG250 associated with a low dose of IL2 for 11 weeks presented durable response in 23% of the patients, with several grade 3 or 4 toxicities [31]. When associated with interferon-alpha (IFNα), cG250 has the most clinical benefit, with complete and partial remissions, 30 months median for overall survival, and 57% of the patients alive after two years. However, almost half of patients developed grade 3 or 4 AEs [34]. The administration of IFNα and especially IFNγ induced CAIX expression in a dose-dependent way in RCC cells, an effect not observed for IL2 [38]. Nevertheless, despite having some therapeutic efficacy, cytokines such as tumor necrosis factor-alpha (TNFα) and IFNγ displayed cytotoxicity to endothelial cells from blood vessels, turning its therapeutic use restricted to locoregional treatments [39]. To reduce toxicity, the association of cG250 fused with a dimeric form of TNFα was tested in a preclinical model of ccRCC, presenting low toxicity and significant antitumor response, with approximately 50% and 60% decrease in tumor size when used alone or in association with IFNγ, respectively [20].

### 2.3. Chimeric Monoclonal Antibody G250 (cG250) Conjugate with Radionuclides

The stability, biodistribution, and therapeutic effect of several radioisotopes conjugated to cG250 alone or with other drugs were tested in RCC, including ^131^I, ^88/90^Y, ^177^Lu, and ^186^Re. In vivo studies in mice with human RCC xenografts treated with ^177^Lu-SCN-Bz-DTPA cG250 yielded the most outstanding results, duplicating the median survival compared to control [19]. The safety of cG250 conjugated with ^131^I was evaluated in metastatic RCC patients, and the dose of 2220 MBq/m^2^ induced only grade I adverse effects without hepatic toxicity [29]. Posteriorly, ^131^I cG250 associated with IL2 was tested, with low grade 3 or 4 AE, but no complete or partial response was observed [32]. The cG250 antibody conjugated with ^177^Lu-SCN-Bz-DTPA and ^177^Lu-DOTA led to higher radiation doses into the tumor, 87 and 78%, respectively. These data associated with preclinical data using the same therapies suggested that these radionuclides were possibly better candidates for radioimmunotherapy than ^131^I-cG250 [15,19]. A phase I clinical trial determined a MTD of 2405 MBq/m^2^ to multi-infusions of ^177^Lu-cG250 since higher doses induced myelotoxicity, with 74% of the patients presenting stable disease three months after the treatment [35]. In phase II, fourteen patients with progressive metastatic ccRCC received ^177^Lu-cG250, and after the first dose, nine of the fourteen patients (64%) had a response, defined as at least stable disease, three months after the treatment. However, most patients developed grade 3–4 thrombocytopenia, leukocytopenia, and neutropenia, and three patients were excluded from the study due to prolonged myelotoxicity. The six remaining patients were selected to repeat the treatment with 75% of the previous dose, and durable responses were achieved in five, with a slow recovery from myelotoxicity in all these patients. The median progression-free survival (PFS) was 8.1 months considering all treated patients [36].

### 2.4. cG250 and Other Associations

The ^125^I-cG250 antibody was tested preclinically with three different types of tyrosine kinase inhibitors (TKI): sorafenib, sunitinib, and vandetanib in mice inoculated with NU-12 RCC cells. Best results were obtained when mice received a TKI daily for 14 days with ^125^I-cG250 infected intravenously in the middle point (seventh day). Vandetanib promoted the most effective association, followed by the groups treated with sunitinib and sorafenib, all compared to ^125^I-cG250 associated with vehicle only [22]. An antibody-uptake hindering occurs after the end of antiangiogenic therapy, limiting the association schemes [40,41]. Another study showed the combination of ^111^In-cG250 injected three days after administration of 40–50 mg/kg of sunitinib for 13 days to treat human RCC engrafted in mice, reducing in 60% the tumor growth compared to the group treated with ^111^In-cG250 alone [25].

### 2.5. Other Antibodies

Display libraries were further used to select new anti-CAIX antibodies with a therapeutic focus. Two selected anti-CAIX mAbs were reported by Ahlskog et al. (2009), named A3 and CC7, presenting high CAIX affinity [42]. However, we have not found published articles that demonstrate the antitumor efficacy of these antibodies.

Xu et al. (2010) questioned if antibodies selected against other CAIX epitopes could be more effective than G250 to recruit effector cells to the tumor site, antagonizing the proliferative effects and CAIX-mediated transformation. Researchers developed an anti-CAIX high affinity human monoclonal antibody panel and tested it against RCC to address this issue. Of all forty antibodies tested, only six exhibited different degrees of effectiveness by inducing surface-expressed CAIX internalization. The antibodies G119 and G36 allowed the internalization of CAIX in endosomes; G6, G39, G37, and G125 showed inhibition of CAIX activity of 40–50% [43]. Chang et al. (2015) tested the antitumor activity of some of these human anti-CAIX antibodies on ccRCC lines in vitro, including SK-RC-09 (high CAIX expression), SK-RC-52 (moderate CAIX expression), and SK-RC-59 (originally negative for CAIX). All monoclonal antibodies limited the migration of ccRCC cells, with G37 inducing the lowest percentage of migration, followed by G119 with almost the same rate of migration, classified as high and moderate, respectively, by the authors. In vivo tests in an orthotopic human ccRCC xenografts model indicated that G37 and G119 reduced tumor weight by 85% and tumor volume by 75%, the most outstanding results observed preclinically with an antibody used alone [24].

Studies conducted by Zatovicova et al. (2012) allowed the development of murine monoclonal antibodies directed to the catalytic site of CAIX, including the mAb VII/20 capable of efficiently inducing receptor-mediated internalization [21]. This antibody was tested in human colorectal carcinoma xenografts in mice ten days after the establishment of the tumor, resulting in similar tumor volume reduction observed for mG250 in the same cancer model [21]. Another mAbs, called chKM4927 and chKM4927_N297D, were tested in a preclinical study of ccRCC, demonstrating a 60% reduction in tumor volume after 32 days of treatment [26].

Petrul et al. (2012) studied high-affinity anti-CAIX mAbs, selected by panning a MorphoSys HuCAL GOLD^®^ library of human (Fabs) fragments against a recombinant ectodomain of CAIX. The BAY 79-4620 mAb anti-CAIX was identified and conjugated to monomethyl auristatin E through an enzyme-cleavable linker and tested in preclinical models of different tumor types. This treatment demonstrated that CAIX-positive human xenografts representing colorectal, gastric, and patient-derived xenografts (PDX) of non-small cell lung cancer (NSCLC) exhibited up to 100% of complete response rate at higher doses [23]. Anti-CAIX antibodies were also expressed on the surface of liposomes containing encapsulated triptolide (TLP). They significantly increased the cellular uptake of TPL, improving its cytotoxic action and duplicating median survival in mice implanted with lung cancer cells [27].

### 2.6. Fusion Proteins

De Luca et al. (2019) reported the characterization of fusion proteins targeting CAIX while simultaneously linked to IL2 and a low-potency TNF mutant (mut). Mice implanted with CAIX positive murine colon adenocarcinoma cells CT-26 treated with the fusion protein IL2-Anti-CAIX(XE114)-TNFmut and IL2-Anti-CAIX(F8)-TNFmut showed around 60% reduction in tumor volume compared to the control group injected with PBS after 18 days of treatment [28].

## 3. Preclinical and Clinical Studies with Anti-CAIX Chimeric Antigen Receptors (CAR) T or NK Cells

As shown in Figure 1, there are diverse generations of chimeric antigen receptors (CAR), which vary according to the extracellular, transmembrane, and intracellular co-stimulatory domains and the ability to secrete bioactive molecules such as cytokines or antibodies. The CAR is usually expressed in T cells or NK cells, directing the immune system to fight against the tumor [44].

Lamers et al. (2006) carried out the first clinical study to verify the safety of a first-generation CD4TM-γ CAR containing scFv developed based on the murine antibody anti-CAIX G250 expressed on the surface of primary human T cells. The clinical protocol included three patients with a CAR T containing mG250 scFv in an intravenous dose escalation protocol consisting of two cycles. The first consisted of 2 × 10^7^ CAR T cells at day 1, 2 × 10^8^ at day 2, and 2 × 10^9^ at days 3–5. Patients were injected with 2 × 10^9^ cells in the second cycle at days 17–19. All patients received IL2 subcutaneously twice daily at days 1–10 and 17–26. Liver enzyme disorders reaching grades 2 to 4 occurred after five infusions, leading to interruption of the treatment in patients 1 and 3. Treatment with corticosteroids was applied to patient 1, and a dose reduction to 2 × 10^8^ cells was used in patient 2. After treatment, patients had disease progression between days 36 and 106. Hepatotoxicity was found due to the CAR T cell attack of CAIX-positive epithelial cells from the biliary duct [45,46]. In a subsequent study using the same first-generation anti-CAIX CAR T cells, eleven patients were divided into three groups: the same first group described in the work mentioned above, and a second group containing five patients treated with 1 × 10^8^ CAR T cells in a conventional phase I clinical strategy, with a maximum of ten CAR T cells infusions on days 1–5 and 29–33 in association with IL2 (5 × 10^5^ IU/m^2^) twice daily on days 1–10 and 29–38. The dose of 1 × 10^8^ CAR T cells induced hepatotoxicity grade 3 in two of five patients, after 10 and 3 infusions, respectively. HACA was present in most patients from groups one and two. In the third group, three patients were treated similarly to the patients in the second group, but with the addition of a strategy to block CAIX recognition in normal tissues: an extra intravenous infusion of 5 mg anti-CAIX cG250 mAb three days before the start of CAR T cell infusions, leaving only CAIX expressed at higher levels in the tumor site available for the CAR T cell action. CAR T cells were detectable in all patients in the second and third groups after the first series of infusions (days 1–5) and persisted until day 29, when the second course of treatment was started (days 29–33). After treatment, cells were detectable for 2–18 days for the second group and 18–34 days for the third. No interruption of the treatment was necessary for the third group of patients, and no HACA was observed. No clinical response was obtained despite the lower toxicity in the third group [47,48]. Another study involving nine patients with metastatic RCC was performed to determine the MTD for the same anti-CAIX CAR T cells using the cG250 monoclonal antibody as a pretreatment strategy to reduce toxicity, allowing the injection of higher CAR T cell doses. Despite the absence of hepatotoxicity, no effective antitumor response was found even with the highest anti-CAIX CAR T dose applied in the patients [47,48]. The authors pointed out the expression of immunogenic γ-retroviral vector-encoded epitopes as the possible cause of the lack of persistence and absence of therapeutic efficiency of these CAR T cells [48]. Some changes in the CAR T cell culture conditions in vitro were tested to improve the effectiveness of this first-generation anti-CAIX CAR T cell therapy. Their findings suggested that anti-CD3/CD28 mAbs with IL15 and IL21 from the onset of T cell activation induced CAR T with increased CAR expression and functionality, with a high proportion of CD8+ T cells and a lower proportion of CD4+ CD25+ CD127- T cells [49].

Another group developed and compared two generations of humanized anti-CAIX CARs based in the scFvG36 in a preclinical setting: a first-generation CD8 CAR, with scFvG36 linked to CD8, truncated extracellular domains, hinge and transmembrane plus TCRζ signaling domain (G36-CD8z), and a CD28 CAR from second generation consisting of scFv G36 fused to CD28 plus TCRζ signaling domain (G36-CD28z). In this study, the administration of the second-generation humanized anti-CAIX CAR T cells containing the CD28 co-stimulatory domain proved antitumor superiority against the first generation construct [50].

Suarez et al. (2016) improved the efficacy of the second-generation anti-CAIX G36-CD28z CAR T cells using a bicistronic lentivector capable of expressing an immune checkpoint inhibitor and the antiprogrammed cell death ligand-1 (PD-L1) IgG1 or IgG4 monoclonal antibody (Clone 42) in a second cassette, in addition to the anti-CAIX CAR expressed by the first cassette. This research was pioneering work on the expression of immune checkpoint blockade antibodies by CAR T cells. The generated lentiviruses were transduced only into CD8 T cells cultured in the presence of IL21, improving the proliferation of CAR T cells when compared to IL2 and maintaining the cytotoxic activity specifically for CAIX-positive ccRCC cells. An orthotopic model of ccRCC in NSG mice was established, and the group treated with anti-CAIX CAR T secreting anti-PD-L1 had tumors five times smaller than the control groups. Additionally, the tumor-infiltrating lymphocytes presented a 50% reduction in the expression of exhaustion markers LAG-3, TIM-3, and PD-1. Anti-CAIX CAR T cells secreting anti-PD-L1 IgG1 induced ADCC when mice were incubated with natural killer (NK) cells [51], with perspectives of improved results in humans, since the NSG model has limitations due to its inherent immunosuppression.

Another exciting strategy combined bortezomib—a proteasome inhibitor for treating relapsed multiple myeloma—with a CAIX-specific third-generation CAR-NK-92 (CAIX-CAR-NK92), consisting of an scFv G250, hinge CD8 and transmembrane regions, and intracellular signaling domains of CD28, associated with the intracellular domains of CD137 and CD3ζ. Ketr-3 and OSRC-2 cells were treated with bortezomib for 24 h in order to test the cytotoxicity of CAR-NK92 and NK92 alone. A xenograft model was performed by subcutaneous injection of the human kidney cancer cell line Ketr-3 expressing luciferase in NOD/SCID mice. After five days, bortezomib was applied intraperitoneally (5 μg/mouse) followed, after one day, by the injection in the tail vein of 2.5 × 10^6^ Anti-CAIX CAR-NK92 cells, compared to control groups that received only NK92 cells, CAR-NK92 cells without the use of bortezomib and only bortezomib. All groups received daily intraperitoneal injection of IL2. The tumor volume of the group treated with CAR-NK92 + bortezomib was three to four times lower than that of the groups treated with CAR-NK92 or bortezomib alone. The anti-CAIX CAR-NK92 or bortezomib alone were equivalent in their effectiveness, presenting tumors with half of the volume of the mice treated with NK-92 without the CAR [52].

Cui et al. (2019) focused on using a third-generation CD8 hinge, CD28 transmembrane intracellular domain, 4-1BB, CD3ζ-based anti-CAIX scFv (type not described) CAR T cells maintained in culture with IL2 in an orthotopic glioblastoma model by intracranial inoculation with 100,000 U251 cells. A total of 2 × 10^6^ anti-CAIX CAR T cells or mocked-transduced T cells were injected into the tumors after one week. The bioluminescence results showed limited tumor growth and prolonged survival of mice with anti-CAIX CAR T cells compared to controls, resulting in a complete tumor remission in 20% of the mice without tumor recurrence within two months of follow-up. They have also tested a combination of anti-CAIX CAR T with the antivascular endothelial growth factor (VEGF) mAb bevacizumab, obtaining tumors almost three times smaller than those observed in the group treated with these approaches alone [10]. The VEGF receptor tyrosine kinase inhibitor, sunitinib, was also tested in association with anti-CAIX CAR T cells. For this purpose, a mouse antihuman CAIX scFv with a c-myc tag at the N-terminus was fused with the hinge and transmembrane domains of human CD8α and cytoplasmic regions of 4-1BB and CD3ζ to construct a second-generation CAR. The CAR T cells were cultured in vitro with IL2, IL7, and IL15. In a lung metastasis model of human RCC in mice, the combination of anti-CAIX CAR T cells with sunitinib resulted in the survival of all mice at the end of the experiments and decreased tumor burden, with improved infiltration and proliferation of CAR T cells followed by 50% reduction in myeloid-derived suppressor cells (MDSCs) infiltration. Enrichment of CD8+ cytotoxic CAR-T cells and less-differentiated stem cell-like memory T cells was observed without harming the proliferation, cytokine release, and cytotoxicity of anti-CAIX CAR T cells, proving the synergistic effects of sunitinib with anti-CAIX CAR T cells against RCC [53].

Recently, anti-CAIX G36 scFv CAR T cells containing 4-1BB as a co-stimulatory domain were tested in a CD4/CD8 ratio of 2:1, leading to complete remission in an orthotopic ccRCC model in NSG mice, which remained tumor-free 72 days after CAR-T cells infusion. This powerful treatment was able to downregulate immune checkpoint genes and reduce the differentiation of regulatory T cells [54].

## 4. Discussion

Most clinical trials targeting CAIX for immunotherapy with results available in the literature have used the murine monoclonal antibody (mAb) G250 or humanized derivatives thereof, such as cG250, developed primarily for diagnostic and nontherapeutic purposes and therefore mainly tested in a conjugated manner with radioisotopes. Such studies mainly targeted patients with ccRCC due to the constitutive expression of CAIX found in most of these tumors. Considering such tumors, studies performed with ^177^Lu-DOTA-mG250 showed a significant improvement, almost triplicating the median survival in an animal model of ccRCC compared to control, with a reduction in metastases and disease stabilization. However, toxicity by HAMA limited the continuity of studies with this antibody. On the other hand, cG250 had favorable pharmacokinetics in patients with advanced metastatic ccRCC, with clinical benefit, mainly found when conjugated with ^177^Lu. In preclinical models, ^177^Lu-SCN-Bt-DTPA had the most relevant results, duplicating the median survival of the mice with human ccRCC. Experimental data suggest that multiple administrations of radiolabeled antibodies may have a more significant therapeutic effect than a single infusion [29,33]. Considering the association with cytokines, the combination of cG250 with IFNα showed the best clinical results with fewer side effects. Furthermore, due to the preclinical results observed, there are perspectives for optimizing the efficiency of cG250 in the association of INFγ conjugated to the dimeric form of TNF-α [20]. cG250 is a combination partner of moderate toxicity with potential synergistic antitumor effects when associated with other therapeutic agents.

Other anti-CAIX antibodies selected later with a therapeutic focus showed much more promising preclinical results than G250 and cG250 when used in an unassociated manner, highlighting G119 and G37 with outstanding results for the treatment of ccRCC [24]. Considering conjugated forms, the anti-CAIX mAb conjugated with auristatin E (BAY 79-4620) presented the most potent preclinical antitumoral effects, with complete response observed against several types of CAIX-positive human tumors in mouse xenografts, including NSCLC PDX [23].

In the context of cell immunotherapy, CAIX was one of the first targets that emerged for therapy with CAR T for ccRCC. At that time, researchers had no prior knowledge of the best conditions for performing CAR T therapy, and the first clinical trial performed with daily injections of first-generation murine anti-CAIX CAR T combined with IL-2 had disappointing results in efficacy and toxicity. The patients developed anti-CAR T cell antibodies and immune responses that led to degrees of hepatotoxicity from two to four, and four out of eight patients had to discontinue treatment. T cell infiltration was found near the bile ducts in liver biopsies due to the expression of CAIX at these sites, and no objective response was detected. With current knowledge, failure of this protocol would be expected, as sequential daily doses of IL2 in association with murine CAR T cells induced a massive immune response. This immune response was able to induce hepatotoxicity but could not promote an objective response against a highly proliferating tumor since first-generation CAR T cells have low sustained maintenance in the body. Moreover, the γ-retroviral vector coding for the CAR expressed immunogenic epitopes, which probably contributed to the lack of persistence and absence of therapeutic efficiency of these CAR T cells [48]. Furthermore, the ccRCC is a solid tumor, and the circulation of CAR T cells in the tumor microenvironment is also a challenge. However, the results obtained with second-generation anti-CAIX CAR T cells alone or capable of releasing immune checkpoint blockade were the most promising in the preclinical setting [51,54] and should be tested in clinical trials.

It was noted that the design of an anti-CAIX CAR T clinical trial would have a better chance of obtaining an objective response with lesser effects if it were evaluated: (1) an affinity denatured humanized anti-CAIX CAR T preferentially based in an scFv derived from anti-CAIX mAbs developed with therapeutic intention (not for bioimaging); (2) a second-generation structure for the CAR; (3) injections of anti-CAIX CAR T cells CD4: CD8 2:1, precultured in vitro with IL7 and IL15, which are due to the production of a more central memory phenotype with more significant proliferation, durability, and antitumor efficiency. The injection of IL2 associated with CAR T is no longer used in most studies; and (4) a CAR T cell capable of payload, e.g., secretion of proinflammatory cytokines or immunological checkpoint blocking mAbs, to improve their performance and persistence would be preferential.

## 5. Conclusions

The latest cellular or monoclonal-based immunotherapy strategies developed targeting CAIX have surprising preclinical results, particularly against ccRCC, demonstrating the need to conduct clinical studies that explore the potential of CAIX as a therapeutic target. Given that G250-based therapies were the only immunotherapies targeting CAIX clinically tested, there is still a range of new therapeutic mAbs and second-generation CAR–based cellular products to be tested in clinical studies that may provide us with a new perspective on improving the prognosis of patients with renal and other types of CAIX-positive tumors. Special attention to conditions that limit toxicity to healthy tissues that express CAIX must be applied.

## Figures and Tables

**Figure 1 cancers-14-01392-f001:**
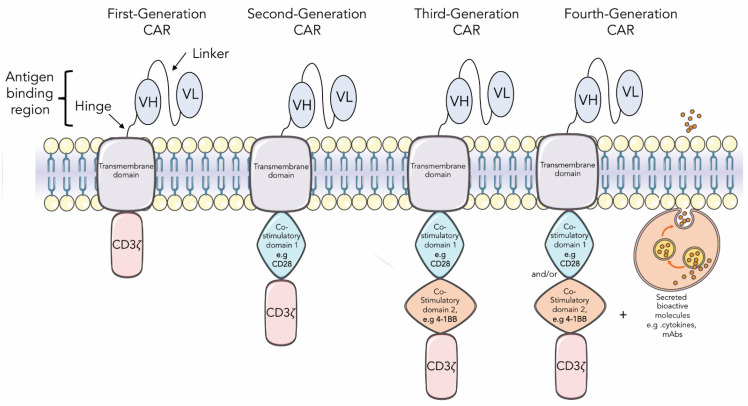
Schematic representation of first, second, third, or fourth generations of chimeric antigen receptors (CAR). CARs are hybrid receptors that comprise an antibody-derived extracellular binding domain selected against a molecular target, usually in the form of a single-chain variable fragment (scFv), and a hinge/transmembrane domain fused to an intracellular signaling domain responsible for activating T cells. First-generation CARs have only one CD3ζ chain in the intracellular domain for activating T cells. Second- and third-generation CARs harbor one and two additional intracellular co-stimulatory domains, respectively. Fourth-generation CARs are CARs of second- or third-generation designed to induce expression of transgenic products constitutively or by induction, such as cytokines or monoclonal antibodies.

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
