# Peer review of "Carbonic Anhydrase IX: A Renewed Target for Cancer Immunotherapy"

_cancers, 2022, doi:10.3390/cancers14061392_

Round 1

Reviewer 1 Report

The paper by Najla Santos Pecheco de Campos and coworkers is an interesting review summarizing different antibodies and cell therapies developed against CA IX. The topic is interesting and the paper is logically organized and well written. Since there is also a huge amount of preclinical and clinical studies on small molecules targeting CA IX, I suggest mentioning this in the introduction adding relevant references

Author Response

Dear Dr., 

We appreciate your comments and suggestions to improve the quality of our paper. Below you will find a point-by-point response to your commentaries

Point#1: "The paper by Najla Santos Pacheco de Campos and coworkers is an interesting review summarizing different antibodies and cell therapies developed against CA IX. The topic is interesting and the paper is logically organized and well written."

Answer 1: Thank you for your valuable feedback. We appreciate it and we are glad to hear that the paper is of value to you.

Point#2: "Since there is also a huge amount of preclinical and clinical studies on small molecules targeting CA IX, I suggest mentioning this in the introduction adding relevant references"

Answer 2: Considering your relevant suggestion, we added the paragraph below to the Introduction, containing general information on small-molecule inhibitors targeting CAIX and referencing some recent reviews and the ongoing Phase Ib/II clinical trial using SLC-0111.

"Several small-molecule inhibitors targeting CAIX were tested in a preclinical setting and recently reviewed[9,10]. An oral sulfonamide SLC-0111 completed a Phase I clinical trial and is currently in Phase Ib/II in combination with intravenous gemcitabine against advanced metastatic pancreatic ductal adenocarcinoma[11]."

Thank you very much, your comments will contribute a lot to the paper.

Best regards,

Prof. Dr. Eloah Rabello Suarez

UFABC/ CCNH

Reviewer 2 Report

The authors in the manuscript entitled "Carbonic anhydrase IX: a renewed target for cancer immunotherapy" reviewed preclinical and clinical data targeting CAIX, a clinically validated target against solid tumors using  strategies based on mAbs, fusion proteins, CAR T cells, and NK cells alone or in combination with cytokines or other immune modulating agents. The manuscript were well written and appropriately described, summarized and cited. The authors can also include a recent article published by Pastorek group (Zatovicova, M., Kajanova, I., Barathova, M. et al. Novel humanized monoclonal antibodies for targeting hypoxic human tumors via two distinct extracellular domains of carbonic anhydrase IX. Cancer Metab 10, 3 (2022)) who described a new set of humanized anti-CAIX antibodies in this manuscript.  

Author Response

Dear Dr., 

We appreciate your comments and suggestions to improve the quality of our paper. Below you will find a point-by-point response to your commentaries

Point#1: "The authors in the manuscript entitled "Carbonic anhydrase IX: a renewed target for cancer immunotherapy" reviewed preclinical and clinical data targeting CAIX, a clinically validated target against solid tumors using strategies based on mAbs, fusion proteins, CAR T cells, and NK cells alone or in combination with cytokines or other immune-modulating agents. The manuscript was well written and appropriately described, summarized, and cited."

Answer 1: Thank you for your valuable feedback. We appreciate it and we are glad to hear that the paper is of value to you.

Point#2: "The authors can also include a recent article published by Pastorek group (Zatovicova, M., Kajanova, I., Barathova, M. et al. Novel humanized monoclonal antibodies for targeting hypoxic human tumors via two distinct extracellular domains of carbonic anhydrase IX. Cancer Metab 10, 3 (2022)) who described a new set of humanized anti-CAIX antibodies in this manuscript."

Answer 2: Considering your relevant suggestion, we summarized the data of Zatovicova, M., 2022 in Table I and added the paragraph below to the section "2.5. Other antibodies".

"Recently data was published with a new murine mAb called IV/18 targeting the sequential epitope of the proteoglycan domain, blocking cell-matrix adhesion without inducing CAIX internalization. This mAb was administered alone compared to VII/20, and both diminished melanoma weight by 60-70% in mice xenografts. However, the humanized counterpart of VII/20, called CA9-hu-1 (HC4LC4), induced higher ADCC levels than the respective humanized counterpart of IV/18 (CA9-hu-2) for breast, cervical, and glioblastoma cancer cells[32]."

Thank you very much, your comments will contribute a lot to the paper.

Best regards,

Prof. Dr. Eloah Rabello Suarez

UFABC/ CCNH